# Evaluation of Ocular Surface after Cataract Surgery—A Prospective Study

**DOI:** 10.3390/jcm11154562

**Published:** 2022-08-04

**Authors:** Agne Sidaraite, Lina Mikalauskiene, Andrzej Grzybowski, Reda Zemaitiene

**Affiliations:** 1Department of Ophthalmology, Medical Academy, Lithuanian University of Health Sciences, 44037 Kaunas, Lithuania; 2Department of Ophthalmology, University of Warmia and Mazury, 10-719 Olsztyn, Poland; 3Institute for Research in Ophthalmology, Foundation for Ophthalmology Development, 61-553 Poznan, Poland

**Keywords:** dry eye disease, ocular surface dysfunction, cataract surgery, phacoemulsification, corneal nerve density

## Abstract

This study evaluated tear film and ocular surface parameters in patients after cataract surgery. Methods: a prospective clinical study included 48 eyes of 48 patients who underwent uncomplicated phacoemulsification performed by one surgeon. Tear break-up time (TBUT), Schirmer’s I test, Cochet–Bonnet esthesiometry and in vivo laser scanning confocal microscopy was carried out and the OSDI questionnaire was filled out. All tests were assessed before and 1 month after the surgery. Results: there were 32 (66.7%) women; 16 (33.3%) men, mean age was 74.08 ± 5.37. Mean TBUT at baseline was 8.6 ± 1.9 s, after the surgery, it was 7.7 ± 2.7 s, *p* = 0.004. Schirmer’s I test was 8.7 ± 3.9 mm versus 8.1 ± 3.7 mm, *p* = 0.002. Mean corneal nerve density at baseline was 15.70 ± 2.34 and at the first postoperative month 14.94 ± 1.48 mm/mm^2^, *p* = 0.02. The OSDI questionnaire score increased from 12.15 ± 10.34 before the surgery to 13.79 ± 10.88 in the first postoperative month, *p* = 0.001. Conclusions: the ocular surface was affected 1 month after the cataract surgery: TBUT was shorter, Schirmer’s I test and corneal nerve density were decreased while the OSDI score increased.

## 1. Introduction

Ocular surface disease is a disorder of the conjunctival and corneal epithelium, lacrimal glands, and meibomian glands. It can be caused by many conditions and can result in deficient or inappropriate tear production which leads to ocular discomfort, pain and even decreased visual acuity [1,2,3].

Dry eye disease (DED) is one of the ocular surface dysfunction causes whose prevalence varies from 5 to 34% and increases with age. Diagnosis of DED is made based on symptoms, which could be evaluated with questionnaires, such as the Ocular Surface Disease Index (OSDI) or the Impact of Dry Eye on Everyday Life (IDEEL) questionnaire and the presence of abnormal test results which includes tear break up time (TBUT), Schirmer’s test, Meibomian gland dysfunction, corneal staining, and others [4,5,6]. DED is now considered an inflammatory disease that could be triggered by environmental or other factors, such as collagen vascular disease, androgen deficiency, refractive surgery, and many others [7,8]. Corneal in vivo confocal microscopy (IVCM) started a new phase in the research of DED pathogenesis because it allowed high-resolution visualization of subbasal corneal nerves and immune cells [9,10,11]. This method showed the morphological changes in cases of DED, such as an increase in the inflammatory dendritic cell density [12,13], and a decrease in the subbasal nerve numbers and their density [14,15], and an increase in the number of beadings and nerve tortuosity [16].

A cataract is mostly an age-related disease whose rate is also increasing with the aging population [17]. The only treatment available is surgery which possibly affects the ocular surface, and it could contribute to developing DED or worsen the symptoms if it is already existent [6,18,19,20]. The Prospective Health Assessment of Cataract Patients’ Ocular Surface (PHACO) study determined that fewer than 25% were diagnosed with DED before the cataract surgery but at least 30% were symptomatic. Moreover, 62.9% had a TBUT ≤5 s and 77% of eyes had positive corneal staining [21]. Sajnani et al. found that persistent post-surgical pain manifesting as dry eye-like symptoms were present in 34% of individuals six months after surgery [22]. According to the American Academy of Ophthalmology, a thorough ocular surface examination is necessary before any kind of ocular surgery to achieve the best possible results regarding patient satisfaction [23].

In this study, we aimed to evaluate some ocular surface parameters before and after the cataract surgery and to determine any possible risk factors and correlations.

## 2. Materials and Methods

The study was carried out in the Department of Ophthalmology, Hospital of Lithuanian University of Health Sciences (LUHS).

The study was conducted in accordance with the Declaration of Helsinki and approved by the Ethics Committee for Biomedical Research of the Lithuanian University of Health Sciences (protocol code BEC-LSMU(R)-40). Informed Consent Statement: Informed consent was obtained from all subjects involved in the study.

A total of 48 eyes (48 patients) having a standard uncomplicated phacoemulsification procedure for age-related cataracts were included in a prospective clinical study and have signed the consent form to participate. The exclusion criteria were concomitant ocular pathologies: any type of ocular inflammation (blepharitis, conjunctivitis, uveitis, and others), glaucoma, ocular allergies, pterygium, other eyelid pathologies (ptosis, dermatochalasis, entropion, ectropion, tumors, congenital abnormalities); recent ocular surgery or trauma (which happened in 2 months period before the cataract surgery); also any systemic diseases and medications which could affect ocular surface (diabetes, connective tissue diseases and infections, also antidepressants, beta blockers and others).

Power calculation was conducted for this study. Calculation of the required minimum sample size of 44 patients was determined by the Paniott formula:$$n=\frac{1}{\Delta^{2}+1/N}$$n—sample size;∆—the size of the sample error (for this study, the size of the sample error of 5% was chosen);N—general whole (it was estimated that per year an average of 50 patients who would fit the inclusion criteria and would not have exclusion criteria are undergoing cataract surgery with this one specific surgeon at the Lithuanian University of Health Sciences Kaunas Clinics).


All 48 patients underwent routine cataract surgery performed by one experienced surgeon (R.Z.). Patients, who experienced any intraoperative complications (such as capsular instability or rupture, vitreous prolapse, etc.) and any complications on the first postoperative day (corneal edema, flare in the anterior chamber or any others) were excluded from the study.

Before the surgery, 5% iodine povidone drops were instilled into the eye. Phacoemulsification was performed with a 2.2 mm clear corneal incision in the superior temporal quadrant if the eye was right and the superior nasal quadrant if the eye was left and two paracenteses. After phacoemulsification, foldable IOL was implanted. At the end of the procedure, an injection of 1 mg of Cefuroxime was conducted in the anterior chamber, and one drop of Levofloxacin was instilled in the conjunctival sac.

Starting from the first postoperative day all patients instilled preserved Levofloxacin eye drops four times a day for 1 week and preserved Dexamethasone eye drops for 1 month starting with four times a day during the first postoperative week and then tapering it by one time a day each week.

All patients underwent the evaluation of the ocular surface before the cataract surgery and in the first postoperative month. The following tests were performed by one investigator in the following order:To evaluate the secretion of tears, the Schirmer’s I test was performed without topical anesthesia. Schirmer paper strips were placed over the lower lid margin, midway between the middle and outer third of the lid. After 5 min strips were taken off and the wet area was measured: the normal secretion in this study was more than 10 mm.Tear break-up time (TBUT) was measured using fluorescein dye. The result was the interval between the last complete blink and the first appearance of a disruption of the tear film. The normal TBUT was ≤10 s.Corneal sensitivity was measured using a Cochet–Bonnet esthesiometer (Luneau; Pruneay-Le-Gillon, France). A nylon filament was placed in the center of the cornea perpendicular to the surface. Patients were asked to look straight ahead and indicate when they felt the stimulus. The result was checked twice before it was noted in the protocol.In vivo confocal microscopy was carried out by the same investigator at the central cornea to evaluate it with laser scanning confocal microscope Heidelberg Retinal Tomograph (HRT II RCM Heidelberg Engineering Inc, Heidelberg, Germany; Rostock Cornea Module). The laser source is a diode laser at a wavelength of 670 nm. The acquired 2D images have a definition of 384 × 384 pixels over an area of 400 × 400 μm. Before the test, one drop of local anesthetic was instilled in the conjunctival sac (Proxymetacaine hydrochloride 0.5%). The single use contact element in sterile packaging (TomoCap) was used during the test which was lubricated with sterile gel (Carbomer 2.5 mg/g) for better image quality. In each case, another independent investigator picked from eight to 10 images which were analyzed with the Image J program (National Institutes of Health, Bethesda, Maryland, USA), Neuron J extension (Biomedical Imaging Group, Lausanne, Switzerland). The evaluation consisted of two parameters: the density of corneal nerves (it showed the length of nerve fibers in one square millimeter, mm/mm^2^) and the number of fibers (measured in one image). Confocal microscopy was conducted for not all the study participants because some patients refused to repeat this test after the cataract surgery, or images were not representable, so such cases (n = 26) were not included in the final analysis.Dry eye symptoms were assessed with a validated OSDI questionnaire [24]. It consists of 12 questions about symptoms and the patient chooses from ‘never’ (0 points) to ‘all the time’ (4 points). The result was calculated using the following formula: OSDI score = (sum of all answered questions) × 100/total number of answered questions) × 4. The OSDI scores range from 0 to 100. Scores from 0 to 12 were considered normal; from 13 to 22—mild dry eye disease; from 23 to 32—moderate dry eye disease; from 33 to 100—severe dry eye disease.

Statistical analysis was performed using SPSS 23.0 software (IBM SPSS, Armonk, NY, USA). The data are presented as absolute numbers with percentages in brackets and as mean and standard deviation (SD) if the distribution was normal and when it was skewed, data are presented as medians with a minimal and maximal worth in the brackets. The distribution of variables was checked using Shapiro–Wilk test. Independent samples were compared using chi-square (χ^2^), Mann–Whitney, or Wilcoxon tests depending on the data distribution. Spearman’s rank correlation coefficient was used to measure the degree of association between two variables. *p* < 0.05 was considered to indicate a statistically significant difference.

## 3. Results

Forty-eight eyes from 48 patients were included in this study: 32 (66.7%) were women and 16 (33.3%) were men. Mean age was 74.08 ± 5.37 years (72.31 ± 5.55 vs. 74.97 ± 5.14 years for men and women, respectively, (*p* = 0.109)). 

### 3.1. OSDI Questionnaire: Results

The preoperative incidence of dry eye disease according to OSDI questionnaire results was 26 (54.2%) while after the surgery it was 31 (64.6%) (Figure 1). There was a significant increase in mean OSDI score after the cataract surgery: from 12.15 ± 10.34 to 13.79 ± 10.88, *p* = 0.001. The postoperative increase in mean OSDI score was noted in both men (from 8.81 ± 7.65 to 10.5 ± 9.34, *p* = 0.054) and women group (from 13.59 ± 10.78 to 15.44 ± 11.35, *p* = 0.008). There was no significant difference in preoperative OSDI scores between men (8.81 ± 7.65) and women (13.59 ± 10.78), *p* = 0.157. Additionally, no correlation between the preoperative OSDI score and patient age was found (r = −0.62), *p* = 0.675.

### 3.2. Corneal Sensitivity

There was no change between preoperative and postoperative corneal esthesiometry results, the median was the same at 5.5 cm (min 4.5 cm, max 6.0 cm) before and after the cataract surgery, (*p* = 0.083). We did not find any corneal sensitivity differences between men and women preoperatively and postoperatively and no correlation between esthesiometry results and age.

### 3.3. Schirmer’s I Test

Normal tear secretion was found in 20 (41.7%) patients while it was decreased for 28 (58.3%) patients before the surgery. In the first postoperative month, diminished tear secretion was found in 29 (60.4%) patients. There was a significant decrease in Schirmer’s I test results after the cataract surgery: from 8.71 ± 3.85 mm to 8.13 ± 3.69 mm, *p* = 0.002. We compared the mean of preoperative and postoperative Schirmer’s test results in both men (from 9.44 ± 5.14 mm to 8.63 ± 4.98 mm, *p* = 0.006) and women group (from 8.34 ± 3.04 mm to 7.88 ± 2.92 mm, *p* = 0.063) the decrease was found in both but only for men it was statistically significant. There was no difference in Schirmer’s test results before the surgery between men (9.44 ±1.28 mm) and women (8.34 ± 0.54 mm), *p* = 0.467.

### 3.4. TBUT

TBUT was shorter than normal for 22 (45.8%) patients before the surgery and for 26 (54.2%) patients in the first postoperative month. Mean TBUT significantly decreased after the cataract surgery: from 8.63 ± 1.89 s to 7.73 ± 2.66 s, *p* = 0.004. There was no difference in mean preoperative TBUT between men (8.75 ± 1.98 s) and women (8.56 ± 1.88 s), *p* = 0.558. After the surgery results tended to decrease for men (from 8.75 ± 1.98 s to 7.5 ± 2.96 s, *p* = 0.20) and women (from 8.56 ± 1.88 s to 7.84 ± 2.53 s, *p* = 0.082) but no significant difference was noticed.

In our study, for 22 (45.8%) patients who had no complaints before the surgery, DED could be diagnosed based on test results: tear secretion was lower than normal for 36.3% and TBUT was shorter than normal for 18.2% of patients.

### 3.5. The Density of Corneal Nerves and the Fiber Number

In vivo confocal microscopy was carried out for 22 (45.8%) patients: 13 (59.1%) women and 9 (40.9%) men. The mean density of subbasal corneal nerves diminished from 15.704 ± 2.339 mm/mm^2^ to 14.942 ± 1.477 mm/mm^2^ after the surgery, *p* = 0.020. In Figure 2 an example of these changes is shown. Postoperative reduction of the mean density of corneal nerves was statistically significant for women (from 15.741 ± 2.763 mm/mm^2^ to 14.857 ± 1.622 mm/mm^2^, *p* = 0.039) but not for men (from 15.649 ± 1.706 mm/mm^2^ to 15.064 ± 1.322 mm/mm^2^, *p* = 0.260). There was no difference in the mean preoperative corneal nerve density between men and women, *p* = 0.556.

The mean number of subbasal corneal nerve fibers was significantly lower after the surgery (from 12.0 ± 2.4 preoperatively to 11.5 ± 2.4 postoperatively), *p* = 0.020. This change noted in both men (from 11.2 ± 2.2 to 11.0 ± 1.9; *p* = 0.480) and women group (from 12.5 ± 2.4 to 11.8 ± 2.7), *p* = 0.070 but no significant differences were noticed. 

We have not found a correlation between the density and the number of subbasal corneal nerves neither before (r = 0.412, *p* = 0.057) nor after the cataract surgery (r = 0.108, *p* = 0.632). There was no correlation between patients’ age and the density (r = −0.395, *p* = 0.069) as well as the number (r = 0.124, *p* = 0.584) of corneal nerves.

No correlation between OSDI score and the density and the number of corneal nerves were found neither before (r = −0.279, *p* = 0.209 and r = −0.152, *p* = 0.499) nor after the cataract surgery (r = −0.134, *p* = 0.553 and r = −0.269, *p* = 0.226).

Positive correlation was found between preoperative corneal esthesiometry results and the number of preoperative corneal fibers (r = 0.449, *p* = 0.036); however, no relation was found after the cataract surgery (r = 0.111, *p* = 0.622). We did not observe a correlation between corneal esthesiometry results and the density of corneal nerves neither before the surgery (r = 0.170, *p* = 0.448) nor in the first postoperative month (r = −0.062, *p* = 0.785).

## 4. Discussion

Our study focused on ocular surface changes after cataract surgery. This topic is becoming more relevant as the prevalence of both cataract and ocular surface disease is increasing with the aging population. 

The incidence of DED before the cataract surgery according to OSDI questionnaire’s results in our study was 54.2%. Gupta et al. reported that 54% of patients before the phacoemulsification had DED [25]. The incidence of DED varies from 22.1 to 100% before the phacoemulsification in studies that determined symptoms using other and/or modified OSDI questionnaires [23,26]. The mean preoperative OSDI score in our study was 12.15 ± 10.34 which was similar to other studies [21,27].

We reported a significant rise in the OSDI score in the first postoperative month when compared with preoperative results and this was also found in other studies [21,28,29,30]. In comparison, Xue and others described a very significant rise in the OSDI score from 12.5 preoperatively to 58.33 in the first postoperative month and 37.5 in the third postoperative month. In the sixth postoperative month, it decreased and became the same as the preoperative result [27]. We found that the mean OSDI score after the cataract surgery increased significantly in the women group, but no significant difference was found in preoperative results between men and women. Although we have not found studies describing OSDI score change after the surgery between men and women, it is noticed that DED is more common in women [5,25,31].

In our study, the mean postoperative TBUT was shorter than the preoperative mean TBUT and this tendency could be seen in the following studies 1 month after the phacoemulsification [27,28,29,30,32,33,34]. A significant change in TBUT was noticed by Kasetsuwan and others: from 12.15 s it decreased to 5.11 s in the first postoperative month and remained short even in the third postoperative month with a mean of 5.21 s [21]. A significant decrease in the first month after the cataract surgery was also noticed in Cetinkaya et al.’s study but in the third postoperative month, TBUT values did not significantly differ from the preoperative results [34]. These studies did not compare TBUT changes in men and women before and after phacoemulsification. In our study, TBUT was lower in the first postoperative month for both men and women but changes were not statistically significant.

We found that before the cataract surgery Schirmer’s I test was less than 10 mm for 28 (58.3%) patients. Moreover, the mean result diminished in the first postoperative month which was also found by other authors in their research [21,27,28,30,32,33,34]. There are a few studies that showed long-term results after the phacoemulsification: Schirmer’s I test results were the same as preoperative baseline 3 months after the cataract surgery in the Cetinkaya and others study [34] and 6 months after the cataract surgery in the Xue and others study [27]. We have not found comparative data of Schirmer’s I test results for men and women before and after the cataract surgery. In our study, we found a decrease in Schirmer’s I test results in the first postoperative month for both men and women but only for men was it statistically significant.

Vehof et al. carried out a large cross-sectional study to find a correlation between DED symptoms and objective test results for men and women. They found that women had higher OSDI scores than men in mild and moderate DED but the correlation between symptoms and overall severity of signs score was significantly lower in women (r = 0.11 vs. r = 0.33 in men, *p* = 0.01) [35]. Li and others tried to find a connection between DED symptoms and sensitivity to pain for both sexes. In women, greater pain sensitivity was found to be associated with higher OSDI and other DED questionnaire scores, whereas in men, depending on the questionnaire, greater pain sensitivity was either not associated with dry eye symptoms or associated with decreased dry eye symptoms [36]. In our study, OSDI scores increased postoperatively in women, although objective test results, such as Schirmer’s test and TBUT did not change significantly.

Prospective studies have shown that corneal nerves are altered in DED [37,38,39,40,41] and after cataract surgery [42,43]. We observed a reduction in the mean density of the subbasal cornea after the surgery, but the mean number of subbasal corneal nerves did not change significantly. To this day, there are only a few studies that focused on corneal nerve changes after phacoemulsification. De Cilla et al. found that the number of corneal nerve fibers diminishes after the cataract surgery in the central cornea and in the temporal cornea (near the incision) they were absent in all cases in the first postoperative month, gradually growing back in the third postoperative month [44]. Misra and others examined patients with diabetes mellitus before and 1 month after the cataract surgery. They found that subbasal nerve density in the central cornea diminished 1 month after the surgery in both the control and diabetes mellitus groups [45]. In our study, we found that postoperative reduction of the mean density of corneal nerves was statistically significant for women, but not for men; the change in the corneal fiber numbers after the phacoemulsification was significant for both men and women.

Our results show that dry eye symptoms and ocular surface dysfunction signs are quite common before the cataract surgery, and they only worsen after phacoemulsification. There are a few studies that evaluated medications for dry eye symptoms and signs occurring after the phacoemulsification: trehalose/sodium hyaluronate [46], trehalose/hyaluronic acid vs. hyaluronic acid [47], 3% diquafosol [48], and 0.05% cyclosporine A [49] which all had positive outcomes.

One study carried out by Favuzza et al. evaluated the effect of a hydroxypropyl guar (HPG) and hyaluronic acid (HA) ophthalmic solution before and during phacoemulsification in terms of post-cataract surgery dry-eye disease (DED) prevention. Patients who had instilled HPG/HA solution three times/day in the preoperative week and for two postoperative months had lower SPEED (Standard Patient Evaluation of Eye Dryness) questionnaire scores and higher tear break-up time scores than the group which only instilled HPG/HA for two postoperative months [50]. Hovanesian et al. determined the effect of topical cyclosporine 0.09% on ocular surface regularity for the patients who were diagnosed with DED before the cataract surgery. Topical cyclosporine 0.09% was prescribed BID for 28 days pre-surgery and this treatment showed a statistically significant improvement in the prediction error of the spherical equivalent outcome of surgery [51]. To our knowledge, only these two earlier described studies evaluated the effect of topical ocular solutions before the cataract surgery to minimize the ocular surface dysfunction after the cataract surgery, so further research is needed to make conclusions about the preoperative and postoperative care after the phacoemulsification.

To our knowledge, this is the first prospective study where TBUT, Schirmer’s test, corneal nerve density and fiber number with OSDI score was compared between men and women after phacoemulsification. However, our study had some limitations, such as a small sample size, short follow-up period after surgery and in vivo confocal microscopy being carried out only in the central cornea. Another limitation of our study is that patients were treated with preserved drops after the surgery so the ocular surface could be affected by it and not only by the phacoemulsification itself. A randomized study regarding this matter could be conducted to find out if the ocular surface is affected less when using preservative-free drops after cataract surgery.

## 5. Conclusions

The ocular surface was affected 1 month after the cataract surgery: TBUT was shorter, Schirmer’s I test’s results and corneal nerve density were decreased while OSDI scores increased. A decrease in Schirmer’s results was found for men while OSDI scores increased for women after the surgery. Corneal nerve density decreased for women while the number of corneal nerves decreased for both sexes. Further research is needed to evaluate if the changes observed in our study are long-term.

## Figures and Tables

**Figure 1 jcm-11-04562-f001:**
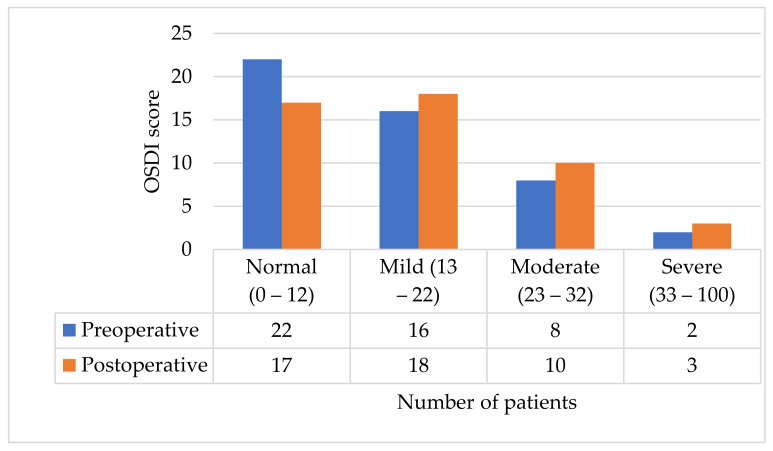
OSDI score before and 1 month after the surgery.

**Figure 2 jcm-11-04562-f002:**
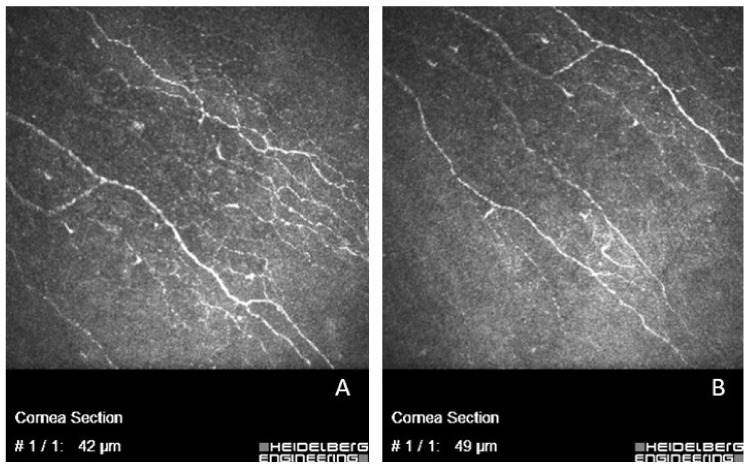
Corneal subbassal nerve plexus before (**A**) and 1 month after the cataract surgery (**B**), the same patient.

## Data Availability

All data supporting reported results can be found in this article.

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
