# Peer review of "Evaluation of Ocular Surface after Cataract Surgery—A Prospective Study"

_jcm, 2022, doi:10.3390/jcm11154562_

Round 1
Reviewer 1 Report
This is an excellent story about the effects of cataract surgery on the ocular surface. A longer term follow up will be most important for the true clinical effects long term from surgery.
It may be helpful to speak about what interventions or treatments may help to reduce the negative impact of surgery on the ocular surface. Since there is a well accepted protocol of antibiotics, steroid and NSAID after surgery, does this study affect the current standard of care? The authors should discuss this more, thanks
Author Response
Dear Sir/ Madam,
We would like to thank you for your review and important comments. We agree that a longer follow-up period is needed to fully evaluate phacoemulsification effect on ocular surface. The following information is added to the discussion section in the manuscript:
Our results show that dry eye symptoms and ocular surface dysfunction signs are quite common before the cataract surgery, and they only worsen after phacoemulsification. There are a few studies which evaluated medications for dry eye symptoms and signs occurring after the phacoemulsification: trehalose/sodium hyaluronate [1], trehalose/hyaluronic acid vs hyaluronic acid [2], 3 % diquafosol [3], and 0.05% cyclosporine A [4] which all had positive outcomes.
One study carried out by Favuzza et al. evaluated effect of a hydroxypropyl guar (HPG) and hyaluronic acid (HA) ophthalmic solution before and during phacoemulsification in terms of post-cataract surgery dry-eye disease (DED) prevention. Patients who had instilled HPG/HA solution 3 times/day in the preoperative week and for two postoperative months had lower SPEED (Standard Patient Evaluation of Eye Dryness) questionnaire scores, and higher tear break-up time scores than the group which only instilled HPG/HA for two postoperative months [5]. Hovanesian et al. determined the effect of topical cyclosporine 0.09% on ocular surface regularity for the patients who were diagnosed with DED before the cataract surgery. Topical cyclosporine 0.09% was prescribed BID for 28 days pre-surgery and this treatment showed a statistically significant improvement in the prediction error of the spherical equivalent outcome of surgery [6]. In our knowledge, only these two earlier described studies evaluated the effect of topical ocular solutions before the cataract surgery to minimize the ocular surface dysfunction after the cataract surgery so further research is needed to make conclusions about the preoperative and postoperative care after the phacoemulsification.
- Cagini C, Torroni G, Mariniello M, Di Lascio G, Martone G, Balestrazzi A. Trehalose/sodium hyaluronate eye drops in post-cataract ocular surface disorders. Int Ophthalmol. 2021;41(9):3065-3071. doi: 10.1007/s10792-021-01869-z.
- Cagini C, Di Lascio G, Torroni G, Mariniello M, Meschini G, Lupidi M, Messina M. Dry eye and inflammation of the ocular surface after cataract surgery: effectiveness of a tear film substitute based on trehalose/hyaluronic acid vs hyaluronic acid to resolve signs and symptoms. J Cataract Refract Surg. 2021;47(11):1430-1435. doi: 10.1097/j.jcrs.0000000000000652.
- Kim S, Shin J, Lee JE. A randomised, prospective study of the effects of 3% diquafosol on ocular surface following cataract surgery. Sci Rep. 2021;11(1):9124. doi: 10.1038/s41598-021-88589-7.
- Kang MS, Shin J, Kwon JM, Huh J, Lee JE. Efficacy of 0.05% cyclosporine A on the lipid layer and meibomian glands after cataract surgery: A randomized, double-masked study. PLoS One. 2021;16(1):e0245329. doi: 10.1371/journal.pone.0245329.
- Favuzza E, Cennamo M, Vicchio L, Giansanti F, Mencucci R. Protecting the Ocular Surface in Cataract Surgery: The Efficacy of the Perioperative Use of a Hydroxypropyl Guar and Hyaluronic Acid Ophthalmic Solution. Clin Ophthalmol. 2020;14:1769-1775. doi: 10.2147/OPTH.S259704.
- Hovanesian JA, Berdy GJ, Epitropoulos A, Holladay JT. Effect of Cyclosporine 0.09% Treatment on Accuracy of Preoperative Biometry and Higher Order Aberrations in Dry Eye Patients Undergoing Cataract Surgery. Clin Ophthalmol. 2021;15:3679-3686. doi: 10.2147/OPTH.S325659.
Reviewer 2 Report
The study aimed to determine the effects of cataract surgery after one month on the ocular surface.
- Why DEQ-5 was not considered in addition to OSDI to determine any underlying dry eye symptoms or history, as DEWS II specifically advises the use of the two questionnaire for diagnosing DED.
- Evidently, there was no power calculation done for the study. How did authors decide to recruit 48 participants in both groups. This is crucial to determine the appropriateness of the study design.
- The effects of cataract surgery have been previously assessed post one and three months. It is unclear why only one month time-point was chosen. Furthermore,
- Why confocal microscopy was not performed on all the participants? As such, the assessment on only 22 participants is insufficient to draw definite conclusions.
- Was only central cornea imaged on IVCM in all the patients?
- How were images selected for processing? Were these randomly selected by the same author who conducted tracing or was it done by a separate investigator in a masked/blinded fashion?
- Was minimally invasive phenol red thread test considered instead of Schirmers test?
- Were osmolarity test (TearLab) or Inflmma dry to assess ocular surface inflammation considered?
- Did authors consider non-contact corneal esthesiometry instead of invasive and non-specific Cochet-bonnet esthesiometer? There are several non-invasive ones are now available worldwide.
- Were all the tests performed by the same clinician?
- Were all the eye assessments performed in the same order? Any disruption due to the first measurement could jeopardize rest of the ocular surface examinations.
- Table 2 shows the differences between different severity of DED and comparative OSDI and TBUT scores. With such small numbers in each group, the p values are seemingly irrelevant.
- A few additional citations are needed:
Iglesias, E., Sajnani, R., Levitt, R. C., Sarantopoulos, C. D., & Galor, A. (2018). Epidemiology of persistent dry eye-like symptoms after cataract surgery: persistent post-surgical pain after cataract surgery. Cornea, 37(7), 893.
Giannaccare, G., Bernabei, F., Pellegrini, M., Guaraldi, F., Turchi, F., Torrazza, C., ... & Vagge, A. (2021). Bilateral morphometric analysis of corneal sub-basal nerve plexus in patients undergoing unilateral cataract surgery: a preliminary in vivo confocal microscopy study. British Journal of Ophthalmology, 105(2), 174-179.
Lum, E., Corbett, M. C., & Murphy, P. J. (2019). Corneal sensitivity after ocular surgery. Eye & Contact Lens, 45(4), 226-237.
Author Response
Dear Sir/ Madam,
We would like to thank you for your review and important comments. Below please find our point-by-point response.
The study aimed to determine the effects of cataract surgery after one month on the ocular surface.
Why DEQ-5 was not considered in addition to OSDI to determine any underlying dry eye symptoms or history, as DEWS II specifically advises the use of the two questionnaire for diagnosing DED.
Answer:
First, this study was carried out in Lithuania and at the moment only OSDI questionnaire was validated in Lithuanian. We also used OSDI to evaluate the dry eye symptoms but not to diagnose the DED. We will take note about this matter and questionnaires in our future studies.
Evidently, there was no power calculation done for the study. How did authors decide to recruit 48 participants in both groups. This is crucial to determine the appropriateness of the study design.
Answer:
Power calculation was done for this study. Calculation of required minimum sample size of 44 patients was determined by the Paniott formula (formula is written in attached file, for some reason we couldn't write it in here).
n – sample size;
∆ – the size of the sample error (for this study, the size of the sample error of 5% was chosen);
N – general whole (it was estimated that per year an average of 50 patients who would fit the inclusion criteria and wouldn’t have exclusion criteria are overgoing cataract surgery for this 1 specific surgeon at Lithuanian University of Health Sciences Kaunas Clinics).
We added a comment on this in Methods section.
The effects of cataract surgery have been previously assessed post one and three months. It is unclear why only one month time-point was chosen.
Answer:
We chose to evaluate patients 1 month after the surgery because this period of follow-up is considered universal in many countries and practises for uncomplicated cases. We mentioned in our manuscript that longer follow-up period is needed to evaluate the phacoemulsification effect on ocular surface, and this will be taken for consideration in the future studies.
Furthermore, Why confocal microscopy was not performed on all the participants? As such, the assessment on only 22 participants is insufficient to draw definite conclusions.
Answer: Confocal microscopy was done for not all the study participants because some patients refused to repeat this test after the cataract surgery, or images were not representable, so such cases were not included in the final analysis.
We added a comment on this in Methodology section.
Was only central cornea imaged on IVCM in all the patients?
Answer: Confocal microscopy was done only in central cornea. We acknowledge that in other studies images are taken near or at the site of the main incision and we will consider this in our future studies.
We added a comment on this in Methods section.
How were images selected for processing? Were these randomly selected by the same author who conducted tracing or was it done by a separate investigator in a masked/blinded fashion?
Answer: The images of all layers of the cornea were obtained by same investigator and the most representative 8 to 10 images were selected for further evaluation by another independent investigator.
We added a comment on this in Methods section.
Was minimally invasive phenol red thread test considered instead of Schirmers test?
Answer: We did consider the Phenol Red Thread test (PRT), but we haven’t found studies which would support that it is superior to the Schirmer test (regarding tear secretion) which is considered a standard. Also, other studies regarding the ocular surface disorder after the cataract surgery also used Schirmer test so in order to compare the results, we needed to carry out the Schirmer test.
Were osmolarity test (TearLab) or Inflmma dry to assess ocular surface inflammation considered?
Answer: We did consider evaluating the ocular surface inflammation with the InflammaDry and TearLab tests but unfortunately, we did not have the equipment and resources to do these tests at the time. We plan to include these tests in our following studies understanding the importance of inflammation in the DED pathogenesis.
Did authors consider non-contact corneal esthesiometry instead of invasive and non-specific Cochet-bonnet esthesiometer? There are several non-invasive ones are now available worldwide.
Answer: We did consider non-contact corneal esthesiometry instead of Cochet-bonnet esthesiometer but unfortunately, we did not have non-contact device, so measurements were done with possessed equipment.
Were all the tests performed by the same clinician?Were all the eye assessments performed in the same order? Any disruption due to the first measurement could jeopardize rest of the ocular surface examinations.
Answer: We confirm that all the tests were performed by the same clinician and all the eye assessments were performed in the same order.
We added a comment on this in Methods section.
Table 2 shows the differences between different severity of DED and comparative OSDI and TBUT scores. With such small numbers in each group, the p values are seemingly irrelevant.
Answer: We do agree that numbers in each group is small but significant differences were found and we would like to show these results.

Round 2
Reviewer 2 Report
Authors have aptly answered most of the comments. However, power calculation still lacks clarity. It is correct 48 patients will be enough to assess changes before and after cataract surgery. However, the comparative analysis between DED – mild, moderate, severe and, control participants cannot provide obvious conclusions with such small numbers. The sub-classification of DED for a comparative analysis needs at least 20 patients in each group.
Author Response
Dear Sir/ Madam,
We would like to thank you for your comment. We agree that sub-classification of DED for a comparative analysis needs at least 20 patients in each group. The information presented in Table 1 was not discussed in the Discussion section and conclusions were not based on it so we decided to exclude this Table and information.